# The emergence of three-dimensional chiral domain walls in polar vortices

Sandhya Susarla [1,2,9,10] ✉, Shanglin Hsu[1,2,10], Fernando Gómez-Ortiz [3], Pablo García-Fernández [3], Benjamin H. Savitzky[1], Sujit Das [4], Piush Behera[5], Javier Junquera [3], Peter Ercius [1], Ramamoorthy Ramesh [1,2,6,7,8] ✉ & Colin Ophus [1] ✉

Chirality or handedness of a material can be used as an order parameter to uncover the emergent electronic properties for quantum information science. Conventionally, chirality is found in naturally occurring biomolecules and magnetic materials. Chirality can be engineered in a topological polar vortex ferroelectric/dielectric system via atomic-scale symmetry-breaking operations. We use four-dimensional scanning transmission electron microscopy (4D-STEM) to map out the topology-driven three-dimensional domain walls, where the handedness of two neighbor topological domains change or remain the same. The nature of the domain walls is governed by the interplay of the local perpendicular (lateral) and parallel (axial) polarization with respect to the tubular vortex structures. Unique symmetry-breaking operations and the finite nature of domain walls result in a triple point formation at the junction of chiral and achiral domain walls. The unconventional nature of the domain walls with triple point pairs may result in unique electrostatic and magnetic properties potentially useful for quantum sensing applications.

Chirality is a unique topological feature that drives many-body interactions in naturally occurring organic molecules and proteins[1], subatomic particle physics[2], and solid-state physics[3]. The symmetries in a chiral system are configured in such a way that its mirror image cannot be superimposed on itself, manifesting a handedness to the system as exemplified by screws and our own hands. Chirality also exists at the microscale/nanoscale level in inorganic and organic materials such as liquid crystals[4], spin textures in ferromagnets[5,6], amino acids, and D/L-glucose molecules[1] with applications in spin selectivity-based quantum sensing[7], non-linear optics[8], and biosensing applications[9]. However, there are very a few examples of chiral inorganic ferroelectric crystals which could have fundamentally different domain textures[10–13]. Over

the past few years, novel polarization textures in ferroelectrics such as merons[14], polar flux-closure domains[15,16], vortices[17], bubble domains[18,19], antivortex[20], three-fold polar vertices[21], super crystals[22,23] and skyrmions[24] have been engineered in oxide superlattices, emerging from the careful interplay of elastic, electrostatic and gradient energies of electric dipoles. The electric dipole arrangement and complex orbital hybridization in these systems have been probed by the X-ray scattering techniques[25], scanning transmission electron microscopy (STEM)[17]-electron energy loss spectroscopy (EELS)[26], phase-field simulations[12,14,15,17,22] and atomistic first- and second-principles calculations[17,18,22,24,25]. Surprisingly, the presence of chirality has been observed in one such topological texture i.e., polar vortices in

[1]National Center for Electron Microscopy, Molecular Foundry, Lawrence Berkeley National Laboratory, Berkeley 94720 CA, USA. [2]Materials Sciences Division, Lawrence Berkeley Laboratory, Berkeley 94720 CA, USA. [3]Departmento de Ciencias de la Tierra y Física de la Materia Condensada, Universidad de Cantabria, Cantabria Campus Internacional Santander, Santander 39005, Spain. [4]Materials Research Centre, Indian Institute of Science, Bangalore 560012 Karnataka, India. [5]Department of Materials Science & Engineering, University of California, Berkeley 94720 CA, USA. [6]Department of Physics, University of California, Berkeley, Berkeley 94720 CA, USA. [7]Department of Physics, Rice University, Houston 77005 TX, USA. [8]Department of Materials Science and Nanoengineering, Houston 77005 TX, USA. [9]Present address: School for Engineering of Matter, Transport, and Energy, Arizona State University, Tempe 85280 AZ, USA. [10]These authors contributed equally: Sandhya Susarla, Shanglin Hsu. ✉e-mail: sandhya.susarla@asu.edu; ramamoorthy.ramesh@rice.edu; cophus@gmail.com

PbTiO₃/SrTiO₃ (PTO/STO) superlattices from resonant soft X-ray scattering (RXS)[25], second harmonic generation (SHG) and second principles calculations[27]. The presence of chirality in polar vortices is an emergent phenomenon because none of the parent compounds such as STO or PTO are known to be chiral. It has been recently shown experimentally and theoretically that the presence of chirality in this system might be due to different sources. First factor, the strongest one, is the coexistence of vortices with an axial component of the polarization, perpendicular to the vortex plane[28]. The second factor is the buckling of the vortices (i.e., a staggered vortex configuration where the center of the clockwise and counterclockwise vortices are located at different heights) combined with different sizes of the up and down domains results in a chiral structure, although its strength is smaller in comparison to the first scenario. This last source of chirality can be reversed by the external electric fields. The first experimental demonstration was in refs. 27,29, where the chirality switches in a reversible, deterministic, and non-destructive fashion over mesoscale regions[27].

Without any prior knowledge, one would expect the as-grown sample as a racemic mixture, i.e., equal amounts of left-handed and right-handed domain enantiomers, where the chirality within each domain comes from a combination of the two sources described above. To have a non-destructive switchable chirality, it is essential to understand the role of the domain walls separating the enantiomers. In other words, what local physical parameters play a role when the handedness in the neighboring domains change? This includes the offset between the centers of the core, the axial component at the centers of the clockwise/counterclockwise vortex, the sense of rotation of the vortices when they merge at the domain wall, the presence of dislocations, or the combined effect of all of them. A proper understanding of how the left/right-handed domains evolve at the nanoscale is crucial to design the new electrically switchable chiral devices that can be measured without scientifically sophisticated techniques. Indeed, a proper answer to this question will pave the way for the use of these chiral textures in next-generation technologies.

Four-dimensional (4D)-STEM can precisely measure strain, and thus spontaneous polarization in ferroelectrics due to the violation of

Friedel's Law[30–32]. This makes 4D-STEM a unique tool to probe polarization in three dimensions and understand emergent chirality in polarization vortices. In the current work, we have used 4D-STEM to probe three-dimensional domain walls in polar vortices in oxide superlattices and understand the nanoscale nature of chirality. We find that both achiral and chiral domain walls coexist in the same system. The chiral to achiral domain wall transition is driven by the change in the local axial and lateral polarization direction across the domain wall. We have discovered a new pair of triple points with the opposite/same sense of rotation at the junction of achiral and chiral domain walls. Finally, we unravel all the possible configurations of chiral and achiral domain walls in this system through different symmetry-breaking operations.

## Results

### Imaging chirality at different length scales

Trilayer (STO)₁₆/(PTO)₁₆/(STO)₁₆ were grown on orthorhombic DyScO₃ (DSO) [110]ₒ substrates with SrRuO₃ as a buffer layer using reflection high-energy electron diffraction (RHEED)-assisted pulsed-laser deposition (PLD) (Fig. 1a). The polar vortex phase in this system is stabilized as a consequence of the interplay between depolarization energy at the PTO/STO interfaces, elastic energy from the tensile strain imposed by the DSO substrate, and the gradient energy in the ferroelectric[17].

A direct way to measure the polar textures in vortex topologies is through electron microscopy using atomic resolution images. We have used different types of STEM and TEM techniques to characterize the exact positions of the vortex centers. Figure 1a shows a low-magnification bright-field STEM image of the superlattice trilayer with an SRO buffer layer along the [1̄10]ₒ zone axis. The diffraction contrast in BF-STEM allows us to directly locate the vortex center as dark contrast; marked using red circles in Fig. 1a. To precisely understand the polarization or displacement texture around each vortex center, we obtained the A-site displacement vector maps at atomic resolution via gaussian fitting at the A-sites ("Methods", Supplementary Information). Figure 1b and Fig. S1 shows the High angle annular dark field (HAADF)-STEM image of trilayer STO/PTO/STO along [1̄10]ₒ

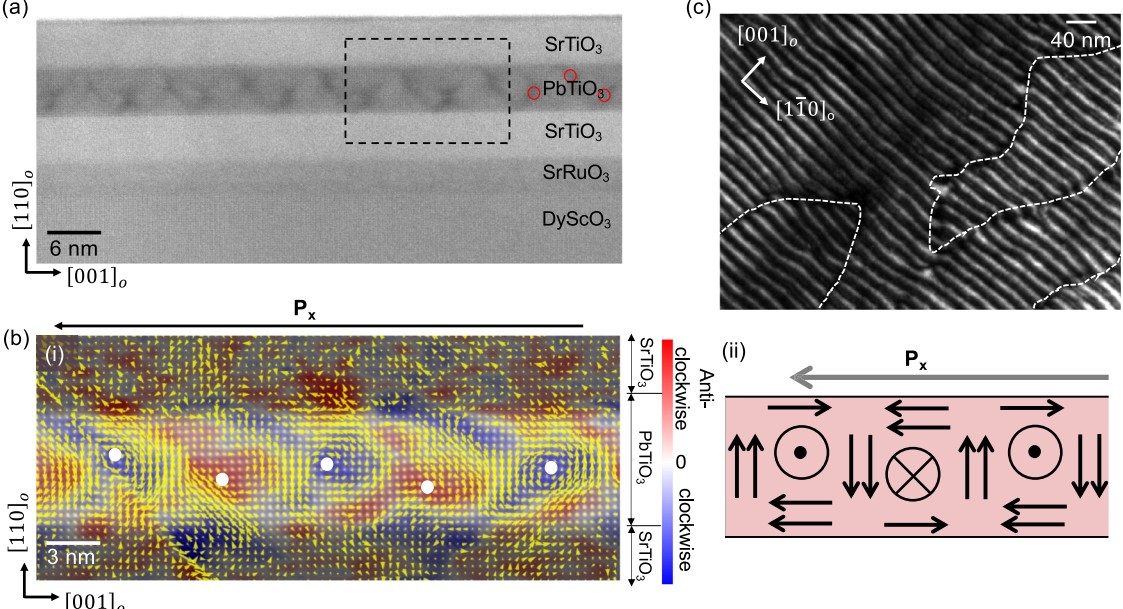

**Fig. 1 | Chirality at different length scales. a** Bright-Field STEM image showing the trilayer stack of 16 STO/16 PTO/16 STO with SRO buffer layer on DSO substrate. The red regions near the center of the PTO layer indicate the position of the vortex core approximately. **b** (i) Vector mapping of the local displacements of A sites of highlighted region in (**a**) overlaid on HAADF-STEM image. The local red- and blue

contrast at the center of the PTO layer indicates the local non-zero curl of displacement. **b**(i–ii) The net lateral polarization resulting from vortex off-centering is indicated at the top. **c** Dark field TEM image along [110]ₒ direction displaying a long tubular vortex structure with domain walls shown as white dashed lines.

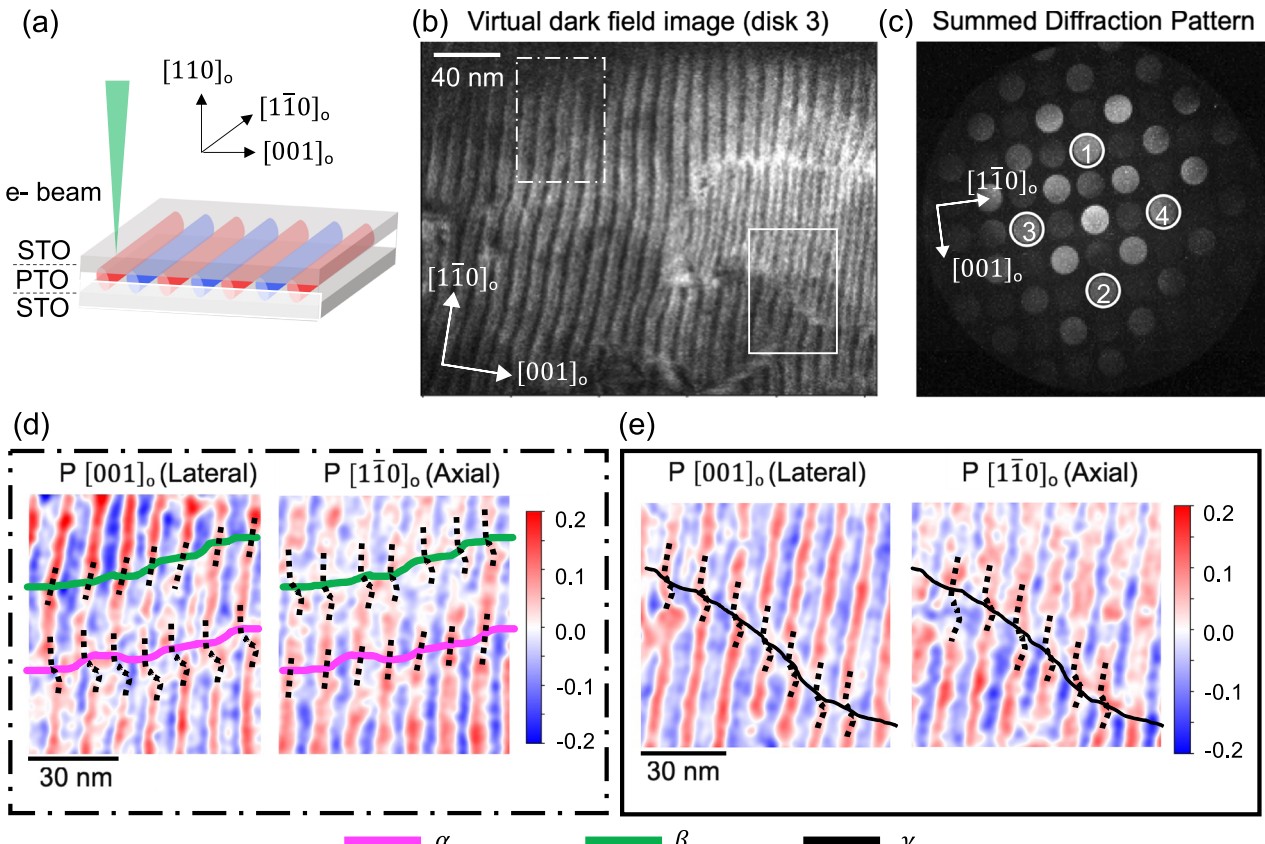

**Fig. 2 | Identification of polarization using 4D-STEM mapping. a** Schematic showing the e-beam scanned in 4D STEM mode across the vortex sample in the in-plane geometry. Most of the signal is coming from the top half of the PTO layer due to the strong scattering by the Pb atomic columns. **b** Virtual dark field image of the vortex region obtained via integrating the intensity of disk 3 from the summed diffraction pattern in (**c**). Zoomed-in images of the **d** dash-dot region and **e** solid line region showing the lateral (P $[001]_o$) and the axial (P $[1\bar{1}0]_o$) polarization maps in vortices. The $\alpha$, $\beta$, and $\gamma$ domain walls can be identified. The $\alpha$ domain wall (magenta curve) has an anti-parallel lateral (P $[001]_o$) component, the $\beta$ (green curve) domain wall has an anti-parallel axial component (P $[1\bar{1}0]_o$), the $\gamma$ domain wall has both axial and lateral components anti-parallel across the domain wall.

zone axis where the brighter regions are PTO and the darker regions are STO. Overlaid yellow arrows show the clockwise and counter-clockwise rotating curls in the displacement of the A-cation in PTO/STO superlattices. The coexistence of the concomitant non-zero curl of polarization (red/blue contrast) with an alternating axial component of the polarization (perpendicular to the plane defined by the vortices) is the first symmetry-breaking operation that results in the emergent chirality in an otherwise non-chiral system. The non-zero curl is larger in continuously rotating polarization textures such as polar vortices[27,30], merons[33], and skyrmions[24] than in other polarization textures such as flux closure domains[15,16] where the curl vanished in the central regions with $180^0$ domain walls. In addition, we observe that the cores of the polarization curls (indicated as blue/red contrast) are not located exactly at the center of the PTO layer, but follow a zig-zag type pattern, giving rise to a net in-plane polarization rotation along $[001]_o$ (lateral component) indicated as $P_x$ in Fig. 1b(i–ii) and Figure S2. This buckling, combined with a small difference in the size of the up and down domains, is the second symmetry-breaking operation that results in a net chirality; in agreement with previous observations[27].

**Identification of different achiral and chiral domain walls**
The various permutations and combinations of atomic scale symmetry-breaking operations such as, a non-zero curl of polarization together with the presence of an axial component, and the buckling of the vortices that yield a non-zero polarization component along $[001]_o$, result in different types of domains walls at the mesoscale. We can visualize the mesoscale domain walls in these topological structures by

imaging the vortices along the $[110]_o$ zone axis using weak beam dark field (WBDF) TEM (Fig. 1c). We can observe the tubular nature of vortex textures by long bright and dark stripes regions in the image. In addition, we observe different domain wall features (indicated as white dashed lines) cutting across vortex tubes. Overall, if we combine our observations from WBDF-TEM and HR-STEM, we observe a three-dimensional vortex structure in the PTO layer sandwiched between two STO layers where the polarization curls follow a tubular pattern (Fig. 2a). Unfortunately, the images formed by WBDF-TEM, HAADF-STEM, and BF-STEM are merely atomic projections and cannot give us an estimate of the physical quantities such as polarization and chirality. On the other hand, 4D-STEM allows us to collect a diffraction pattern at each probe position, which can then be used to create precise maps of the physical quantities such as strain and polarization. The diffraction pattern in 4D-STEM offers a unique advantage over HAADF-STEM in polar vortices because direction of polarization can be accurately measured. For the present experiment, we performed 4D-STEM imaging on a trilayer STO/PTO/STO along the $[110]_o$ zone axis (Fig. 2a). We used a probe size of ~7 Å, larger than the STO/PTO unit cell dimensions (~4 Å) to remove the atomic-scale signal and to estimate the polarization at unit cell length scale. The 4D-STEM analysis was carried out using an open source py4DSTEM analysis package[34]. We define the $[1\bar{1}0]_o$ direction as axial and the $[001]_o$ direction as lateral. The rotation calibration was performed between the real space and the diffraction space images to determine the lateral and the axial zone axis. Details are given in the supplementary information. For the initial visualization of the polar textures, we created a virtual dark field image using the

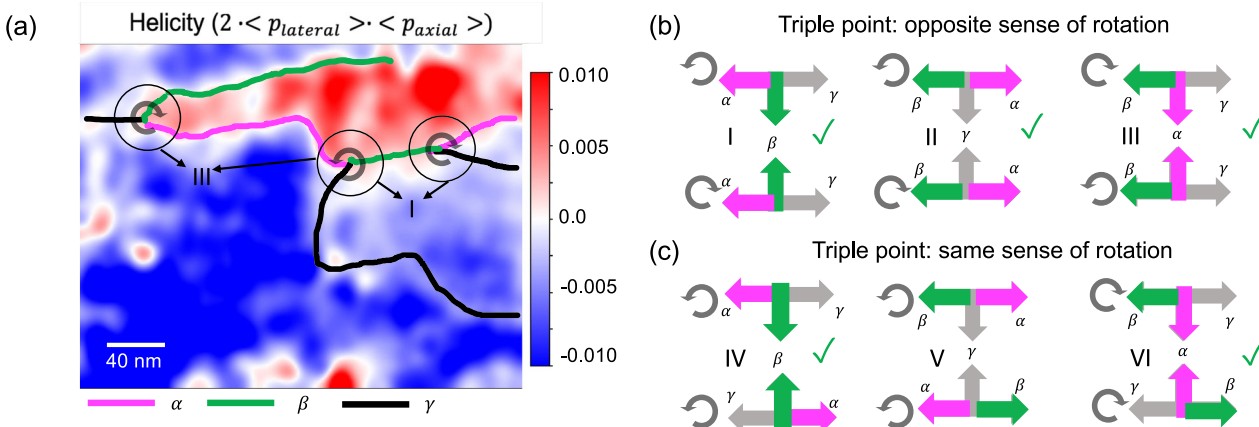

**Fig. 3 | Identification of triple point topologies and chiral domain walls. a** The helicity map formed from Fig. 2b shows left and right-handed domains separated by $\alpha$ (magenta) and $\beta$ (green). In addition, achiral domain walls (black line) also coexist. The resultant triple point topologies formed due to the coexistence of chiral and achiral domain walls are shown in encircled regions. The sense of rotation of these triple point topologies is indicated in the encircled region. **b**, **c** Possible pairs of triple points. **b** Triple points with opposite sense of rotation with the point of inversion along $\beta$, $\gamma$, and $\alpha$, and **c** represents triple points with the same sense of rotation. The green ticks on the side shows what has been observed in the present experiments.

[2$\bar{1}$0]$_o$ disk (disk 3) as shown in Fig. 2b, c (More dark images in Figure S4). The polarization from the PTO layer can be determined qualitatively by subtracting the intensity of the opposite Friedel pair disks due to the violation of Friedel's law[30–32]. This polarization mapping is slightly different from strain mapping. In the former, intensity of the opposite freidel pairs disks are used whereas in the latter the precise position of the diffracted disks is determined[35]. The method for subtracting the intensities of opposite Friedel's pair disks would also work for analogous Pb$_x$Zr$_{1-x}$TiO$_3$ (PZT) where polarization is suppressed under large tetragonality[36]. The (hkl) Friedel pairs were chosen along [001]$_o$ and [1$\bar{1}$0]$_o$ directions to determine the pure lateral and the pure axial polarization, respectively. We note that the other (hkl) directions may also break the Friedel's law, but the associated polarization will be a combination of the lateral and the axial component. The polarization maps corresponding to regions delimited by the rectangles in Fig. 2b are shown in Fig. 2d, e. We observe alternate longitudinal red and blue stripes representing the positive and negative polarization in both the lateral ([001]$_o$) and the axial [1$\bar{1}$0]$_o$ directions together with the domain walls as seen from the WBDF-TEM images marked with green and magenta lines in Fig. 2d and with a black line in Fig. 2e. We observe that the axial polarization magnitude is relatively smaller than the lateral polarization in agreement with the predictions from previous second principles calculations[25,27].

We track the relative domain shift across the wall by the black dotted line as shown in Fig. 2d, e. In Fig. 2d, we observe an $\alpha$-domain wall (magenta line) where the lateral polarization shifts whereas the axial polarization remains the same, as shown in Fig. 3. We also detect a $\beta$-domain wall (green line), where the axial polarization shifts, but the lateral polarization remains the same. Finally, in Fig. 2e there is a third domain wall configuration i.e., a $\gamma$-domain wall (black line) as well where both the lateral and the axial polarization shift. We verified this observation with the average line profiles across the domain walls (Supplementary information, Figure S3). To detect whether a change of chirality occurs at these domain walls, a way to quantify the chirality is required. The order parameter that best captures the breakdown of chiral symmetry is the helicity H of the chiral field. In our case, the chiral field is the polarization, and for the helicity, we borrow the definition from fluid dynamics:

$$\mathcal{H} = \int \vec{p} \cdot \left( \vec{\nabla} \times \vec{p} \right) d^3r, \quad (1)$$

where $\vec{p}$ is the local value of polarization. Note that $H$ changes sign upon a mirror symmetry reflection[37,38]. A non-zero helicity means chirality or lack of mirror symmetry of the polarization texture: right (left) handedness can be associated with positive (negative) values of $H$. Assuming a vortex structure where the polarization lines in the plane defined by the vortices are closed, and that we can measure the axial and the lateral components of the polarization at the topmost PTO layer, then the previous equation can be estimated by

$$\mathcal{H} = 2 \cdot <p_{lateral}> \cdot <p_{axial}> \quad (2)$$

where p$_{lateral}$ and p$_{axial}$ are polarization along lateral and axial directions (Supplementary information, Figure S5). Using this equation we can understand the nature of domain walls found in Fig. 2d, e. For the $\alpha/\beta$ domain wall, only one of the lateral/axial polarization sign changes across the domain wall. This causes a change in the overall sign of helicity, thus making them chiral domain walls. On the other hand, the $\gamma$-domain wall has both the lateral and the axial polarization switch, which doesn't change the overall helicity of the system, thus making it an achiral domain wall.

## Identification of triple points

We can use the qualitative lateral and axial polarization data from 4D-STEM and create a map of the helicity over a large scale using the equation 2 as shown in Fig. 3. The red and the blue regions in the chirality maps indicate different signs of helicity in the system, making them left-handed and right-handed chiral domains separated by the $\alpha$ or the $\beta$ domain wall. We find that most of these chiral and achiral domain walls were not visible in the virtual dark field images. Further, we also observe a unique triple point topology at the mesoscale whenever the two types of chiral domain walls meet an achiral domain wall as seen from black-encircled areas, thus forming a quasi-1D defect in the network of chiral and achiral domain walls. These triple points tend to exist in pairs and exhibit a sense of rotation via the transition from $\alpha$ to $\beta$ to $\gamma$ domain wall and vice-versa (Fig. 3, Figs. S3–S4), similar to what has been observed previously in the trimerized domain walls in hexagonal manganates[39], vortex-antivortex phases in intercalated Vander-Waal ferromagnets[6], and ferroelectric vortex cores in BiFeO$_3$[40]. The sense of rotation in a triple point can be the same or opposite depending on the arrangement of $\alpha$, $\beta$, or $\gamma$ domain walls. Figure 3b,c illustrates this situation.

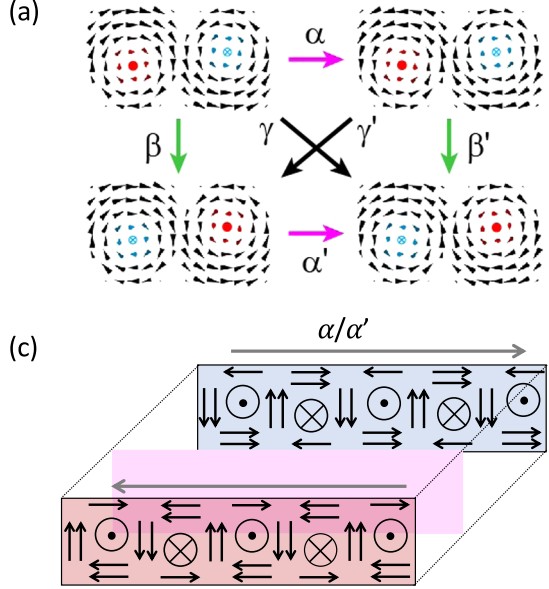

**Fig. 4 | Types of chiral domain walls. a** Four possible combinations of alternating clockwise/counterclockwise vortices are displaced, Top and bottom cartoons differ by the direction of the axial component of the polarization (red dot and blue cross). Left and right cartons differ by the curl of the polarization. The possible domain walls between these configurations are marked as $\alpha/\alpha'$, $\beta/\beta'$, and $\gamma/\gamma'$. $\alpha/\alpha'$ and $\beta/\beta'$ domain walls change the chirality at the domain wall. $\gamma/\gamma'$ domains preserve the chirality. **b–d** Three-dimensional representation of three types of chiral domain walls observed by 4D-STEM measurements.

## Explaining the origin of triple points

Whenever a pair of domain walls ($\beta$, $\gamma$ or $\alpha$, $\gamma$ or $\alpha$, $\beta$) break the inversion symmetry across $\alpha$, $\beta$ or $\gamma$ domains, we form a triple point pair with the opposite sense of rotation. If the inversion symmetry across $\alpha$, $\beta$ or $\gamma$ domains is not broken, we form triple point pairs with the same sense of rotation. The origin of triple point pairs can be understood by the following hypothesis. Consider an example of the first triple point pair in Fig. 3b. If this particular type of triple point has to be isolated, then the $\beta$ boundary would infinitely separate the positive and negative chirality regions. If $\beta$ boundary is not infinite, then it has to meet an $\alpha$ boundary somewhere to continue separating the positive and negative chirality regions. Now, if a $\beta$ boundary (change in axial polarization) meets an $\alpha$ boundary (change in lateral polarization), a $\gamma$ boundary appears (change in both lateral and axial polarization). In such a scenario, we get another triple point in the vicinity of the first triple point thus explaining the origins behind the existance of pairs for the majority of cases. $\alpha/\beta$ boundary could be isolated only in very special circumstances when they are born/die at the surface or they are infinitely long.

From a theoretical perspective, there are three symmetry-breaking operations for the formation of domain walls, (1) Change in the lateral polarization direction, (2) Change in axial polarization direction, and (3) Change in the net lateral polarization due to the vortex core shifting away from the center. If we consider all three factors, then we expect to have six types of domain walls as shown in Fig. 4. The first pair of chiral domain walls, $\alpha/\alpha'$, results from a combination of the factors 1 and 3. The second chiral domain wall pair, $\beta/\beta'$, results from a combination of the factors 2 and 3. The third achiral domain wall pair results from all the three factors. Unfortunately, it is challenging to measure the quantitative net lateral component in the vortices due to the very low sensitivity of electron scattering to sample changes along the beam direction. Due to this, $\alpha/\alpha'$, $\beta/\beta'$, and $\gamma/\gamma'$ are degenerate in this 4D-STEM experiment and thus we observe only three types of domain walls. This has been consistent in multiple such 4D STEM images as observed in Figs. S6–S7. Future studies, such as depth sectioning or sample tilting experiments, may be able to probe this variation[41].

## Discussion

We have unraveled the nanoscale three-dimensional domain wall network in topological polar vortices using quantitative 4D-STEM techniques. The polar vortex oxide superlattice has an emergent chirality through different symmetry-breaking operations in the non-zero curl of polarization along with the alternate axial polarization component and vortex buckling-induced net-in plane lateral polarization component. The interplay of these symmetry-breaking operations result in the formation of two types of chiral and an achiral domain wall within the tubular vortex topologies. Topology-driven domain wall existence in our work is unusual in comparison to other electrostatic conditions-driven domain walls in the ferroic materials[16,39]. The finite nature of the chiral and the achiral domain walls result in the formation of unique triple points whenever these domain walls intersect. The most probable existence of these points is in pairs with the same/different handedness, similar to multiferroic materials such as barium hexaferrite[42] and BiFeO$_3$[40]. To our best understanding, such an unconventional scenario has not been seen yet in improper ferroics literature. We hope that our studies could inspire future experiments to understand the electronic and the magnetic transport at these triple points within the network of chiral and achiral domain walls in polar vortices oxide superlattices.

## Methods

### Synthesis

[(PTO)$_{16}$/(STO)$_{16}$] trilayer with SrRuO$_3$ buffer layer was synthesized on single-crystalline DyScO$_3$ (011) substrates via reflection high-energy electron diffraction (RHEED)-assisted pulsed-laser deposition (KrF laser). The PTO and the STO layers were grown at 610 °C in 100 mTorr oxygen pressure.

### STEM Sample preparation

In-plane [(PTO)$_{16}$/(STO)$_{16}$] trilayer grown on SrRuO$_3$/DyScO$_3$ substrate were mechanically polished using a 0.5° wedge in Allied Multiprep. The samples were subsequently Ar ion milled in a Gatan Precision Ion Milling System, starting from 3.5 keV at 4° down to 1 keV at 1° for the final polish. The HAADF-STEM images were acquired using double

aberration corrected TEAM I microscope operated at 300 kV under non-monochromated mode.

## HR-STEM vector mapping

The vector mapping was performed via Gaussian fitting of A site atomic positions on the drift-corrected HR-STEM images[43]. First, all the A-sites in the drift-corrected images were identified using "Atomap" atom finding tool[44]. Once the atoms were identified, the atomic planes were divided into different zone axis such as along $[001]_O$ and $[110]_O$. The deviation in local A-displacement was found by taking the difference between the local A site displacement and the corresponding average displacement in the local zone axis plane. The displacement vectors were further interpolated into a grid Cartesian grid and then differentiated to obtain strain tensor maps. The infinestimal rotation or the curl of the displacement of vortices was calculated using the following equation:

$$\theta = \frac{1}{2}\left(\frac{\partial u}{\partial y} - \frac{\partial v}{\partial x}\right)$$

The color bar in the curl of displacement plot is plotted with respect to the mean intensities in the PTO layer.

## 4D STEM measurement and analysis

All 4D STEM experiments were carried out on TEAM I microscope (aberration-corrected Thermo Fisher Scientific Titan 80-300) using a Gatan K3 direct detection camera located at the end of a Gatan Continuum imaging filter. The microscope was operated at 300 kV with a probe current of 100 pA. The probe semi-angle used for the measurement was 2 mrad. Diffraction patterns were collected using a step size of 1 nm with 514 by 399 scan positions. The K3 camera was used in full-frame electron counting mode with a binning of 4 pixels by 4 pixels and a camera length of 1.05 mm. The exposure time for each diffraction pattern was 47 ms. The 4D STEM analysis was carried out using the py4DSTEM modules. Briefly, rotation calibration was performed between the diffraction and image plane to identify the right orientation of the zone axis. For that process, the defocused image in the Ronchigram was compared to the focused scan image and the relative orientation of the two images was compared. Once the zone axis was identified, all the disks in the diffraction pattern at each probe position were fitted using the disk fitting function. The so-called polarization maps were generated by taking the normalized intensity difference between the opposite Friedel pair disks. Subsequently, the signal-to-noise in these polarization maps was improved by using a combination of band pass Gaussian filters. By using a high-pass Gaussian filter, we also minimized the dominating thickness contrast.

## Data availability

All data are available in the main text or the supplementary materials. The raw data can be made available upon request.

## Code availability

The code can be made available upon request.

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

## Acknowledgements

All the electron microscopy experiments were carried out at the National Center for Electron Microscopy (NCEM) located in the Molecular Foundry user facility at Lawrence Berkeley National Laboratory. Work at the Molecular Foundry was supported by the Office of Science, Office of Basic Energy Sciences, of the U.S. Department of Energy under Contract No. DE-AC02-05CH11231. S.D. acknowledges Science and Engineering Research Board (SRG/2022/000058) and Indian Institute of Science start-up grant for financial support. S.S., P.B., S.L., and R.R. are supported by the DOE Office of Science, Basic Energy Sciences, Materials Sciences, and Engineering Division under contract DE-AC02-05-CH11231 within the Quantum Materials program (KC2202). C.O. acknowledges support from a DOE Early Career Research Award. F.G.-O., P.G.-F., and J.J. acknowledge financial support from Grant No. PGC2018-096955-B-C41 funded by MCIN/AEI/10.13039/501100011033 and by ERDF "A way of making Europe," by the European Union. F.G.-O. acknowledges financial support from Grant No. FPU18/04661 funded by MCIN/AEI/10.13039/501100011033. B.H.S. was supported by the Toyota Research Institute.

## Author contributions

S.S., R.R., and C.O. conceived the idea, and designed the experiments. S.S. analyzed the 4D STEM datasets, made the figures and wrote the initial draft of the manuscript. S.L.H. performed the 4D STEM experiments. B.H.S. provided inputs for the scripts of the 4D STEM analysis. F.G.O., P.G.F., and J.J. helped in providing intellectual inputs regarding the origin of chiral domain walls and triple points. S.D. and P.B. conducted thin film fabrication and preliminary structural characterization. P.B. and P.E. participated in revising the manuscript. R.R. and C.O. supervised the project.

## Competing interests

The authors declare no competing interests.
