## [Peer Review File · Nature Communications]

The emergence of three-dimensional chiral domain walls in polar vorticesREVIEWER COMMENTS

Reviewer #1 (Remarks to the Author):

This paper investigates the local polarization within tubular vortex topologies in STO/PTO/STO superlattice by 4D-STEM. Through quantifying the helicity based on polarization, they discussed the domain chirality separated by different domain walls. The authors discovered new pair of triple points with the opposite/same rotation at the junction of achiral and chiral domain walls. They further discussed that all possible configuration of three point topologies with reliable analysis of the origin of the triple points. This article presents an interesting study of topologically driven chiral domain walls in oxide superlattices by using specialized electron microscopy. I have some comments:

1. The polarization measurement based on 4D-STEM is an important basis of the article. The authors briefly mentioned in the article that 4D-STEM can precisely measure polarization in ferroelectrics due to the violation of Friedel's Law, I think it is necessary to further elaborate the principle and quantitative details, which is crucial for this article. For example, why choose the difference between 1, 2, 3, and 4 pairs to map the polarization, and can we get the same results by performing similar operations on other pairs?
2. In fact, 4D-STEM only maps the tetragonality (strain). Although it is usually positively correlated with the polarization, but sometimes it is not. For example, at the surface of PZT, the polarization is suppressed although the tetragonality (strain) is larger (Nature Communications | 7:11318 | DOI: 10.1038/ncomms11318). In the vortex system, I would expect this case is more complicated. I wonder how the author handle this. Related discussion is needed to avoid any misleading.
3. Vortex is a three-dimensional polar structure. Obviously, in the STEM image the polarization along the viewing direction is not uniform for this sample, i.e., near the core and far away from the core should have different polarization. For the conventional HAADF-STEM method without the depth resolution, it is very difficult to precisely measure the in-plane polarization to extract the chirality information. However, I don't see how the 4D-STEM solve this problem. Did the authors simply ignore this and extract the averaged polarization information? Detailed discussion is needed.
4. I wonder how to determine the position of vortex cores from the BF-STEM image in Fig. 1a, I can only distinguish the dark contrast of the wave shape. How to correlate the bright/dark contrast with the core positions. Besides, it would be better to remove a few red circles to show more details.
5. Both the virtual dark field image and the actual dark field image in Fig. 1c are constructed from diffraction information. Why are these domain walls invisible in virtual dark field images? In addition, the dark field image in Fig. 1c need to clarify the selected diffraction point.
6. Can different domain walls be distinguished by dark field images? e.g., through branching or stripe periodicity.
7. I found some problems in the cited references in this manuscript. For example, the authors mentioned the "The first experimental demonstration was in 25, where chirality switches...". In fact, at the same time another group published an article of switching chirality of polar vortex at the atomic scale STEM and dark-field TEM, which has been totally ignored in this manuscript. (Sci. China-Phys.Mech. Astron. 65, 237011 (2022), <https://doi.org/10.1007/s11433-021-1820-4>). Another one is the authors mentioned that "novel polarization textures in ferroelectrics such as merons, polar flux-closure domains,

vortices.....in oxide superlattices”, ignoring the representative polar antivortex (Nature Communications | (2021) 12:2054, <https://doi.org/10.1038/s41467-021-22356-0>), and three-fold polar vortices (Nature Communications | (2022) 13:6340, <https://doi.org/10.1038/s41467-022-33973-8>)

8. Some minor suggestions: Fig. 1c lacks the scale bar; Fig. 3a lacks the axis; “Once the atoms were identified, the atomic planes were divided into different zone axis such as along [001]o and [001]o” seems to be a typo here.

Reviewer #2 (Remarks to the Author):

S. Susarla et al. reported the chirality engineering of the topological polar vortex via atomic-scale symmetry-breaking operations. In this work, 4D-TEM results display the topology-driven three-dimensional domain walls, and the manuscript was well organized. As the author said, the chirality of the polar vortex is governed by the perpendicular and parallel polarization of the tubular vortex. In my opinion, polarization vector mapping of the vortex region in in-plane geometry will be a more intuitive way to demonstrate the vortex' chirality. Furthermore, the 4D-TEM data needs to be reanalysed, otherwise it is difficult to support the existing conclusions. Two main concerns are as follows:

1. In Figure 1b, it is difficult for the reader to locate the vortex center, which is consistent with the position marked by the author.
2. The combination of Figure 2 a, 2e and Figure 4e, there is a dislocation structure with respect to the tubular vortex across the γ -domain wall, in accordance with Figure 1b shows. However, in Fig. S3 and S4, the Virtual image has obviously the same boundary (in the middle of the image) as in Fig.2b, but the author does not identify it as any vortex domain wall. Could authors explain this difference? Perhaps, the polarization vector mapping on HAADF-STEM image will illustrate the issue more directly.

Reviewer #3 (Remarks to the Author):

The manuscript by Susarla et al. presents a structural analysis of the 3D domain wall network in topological polar vortices formed in $(\text{SrTiO}_3)_{16}/(\text{PbTiO}_3)_{16}/(\text{SrTiO}_3)_{16}$ trilayers grown on SrRuO₃-buffered DyScO₃ substrates. The microstructure of the films is examined in detail by means 4D-STEM imaging. Thus, lateral and axial polarization maps are obtained by taking the normalized intensity difference between opposite Friedel pair disks. This allows identifying three distinct types of domain wall configurations having different parallel/antiparallel axial/lateral components. Thus, two chiral and one achiral domain walls are identified. Finally, the authors observe that the domain walls meet at triple points, which typically appear in pairs. They hypothesize that these topological defects could lead to unique electrostatic and magnetic properties useful for quantum sensor applications.

The manuscript presents an original and very good experimental and analytical work on the microstructure of domain walls in topological polar vortices. Overall, the manuscript is clearly written and the figures are well elaborated. I enjoyed reading the manuscript, although I suggest introducing a couple of changes to make it a bit more comprehensible. In addition, I am listing also here some minor amendments.

1. In page 4 of the manuscript, "Supplementary Information" should be deleted, as there is no additional information in the SI referring to the trilayer stack.

2. At the end of page 5, it reads that in Figure 1b "a zig-zag type pattern, giving rise to a net in-plane polarization rotation along $[001]_o$ (lateral component) indicated as P_x ". I have difficulties seeing this net in-plane polarization. Could the authors plot the resulting in-plane polarization by averaging it along the horizontal direction and plotting it next to the image? This is not obvious from the figure. It seems to me that the P_x at the top of the PTO film should cancel with the P_x at the bottom of the PTO film.

3. Again, in Figure 1b, it seems there are vortex cores also in the bottom STO layer. Can the authors comment on this?

4. The scale is missing in Figure 1c.

5. It is a bit complicated to follow the discussion of the possible triple points depicted in Figure 3b. Could the authors correlate the triple points observed in Figure 3a (and also in the SI) to the triple point pairs represented in Figure 3b? This would make it easier to follow the explanation.

6. In the caption of Figure 4 it reads "The chirality for each type of domain is indicated by a sketched hand", but in the figure there are no sketched hands. Please, remove the sentence.

7. In the Materials and Methods the authors should indicate how was the sample for S/TEM prepared.

8. In page 14, please change "low-pass and high-pass Gaussian filters" for "band-pass Gaussian filters".

Based on my previous comments I recommend the publication of the manuscript of Susarla et al. in Nat. Commun. after minor revisions.

REVIEWER COMMENTS

Reviewer #1 (Remarks to the Author):

This paper investigates the local polarization within tubular vortex topologies in STO/PTO/STO superlattice by 4D-STEM. Through quantifying the helicity based on polarization, they discussed the domain chirality separated by different domain walls. The authors discovered new pair of triple points with the opposite/same rotation at the junction of achiral and chiral domain walls. They further discussed that all possible configuration of three-point topologies with reliable analysis of the origin of the triple points. This article presents an interesting study of topologically driven chiral domain walls in oxide superlattices by using specialized electron microscopy. I have some comments:

We thank the reviewer for reading our manuscript. We have tried our best to answer the reviewer's questions.

Question 1. The polarization measurement based on 4D-STEM is an important basis of the article. The authors briefly mentioned in the article that 4D-STEM can precisely measure polarization in ferroelectrics due to the violation of Friedel's Law, I think it is necessary to further elaborate the principle and quantitative details, which is crucial for this article. For example, why choose the difference between 1, 2, 3, and 4 pairs to map the polarization, and can we get the same results by performing similar operations on other pairs?

Response: Friedel law states that the diffraction intensities from (hkl) and (\overline{hkl}) planes are similar. However, this rule breaks for non-centrosymmetric materials where the opposite Friedel pair disks have differential intensities due to multiple scattering effects. In this work, we have chosen to use the 1,2, 3 and 4 because these are the primary (hkl) directions along which the effect of Friedel's law breaking is the maximum. The intensity difference between a particular (hkl) Friedel pair is also coupled to the effective polarization along that direction. Since, we were concerned about purely lateral and axial polarization, we choose disks 1, 2, 3 and 4. The other pairs along different directions might have mixed lateral and axial polarization signals and hence it is difficult to interpret the resultant polarization maps.

We have now added a discussion about it in the main manuscript:

The polarization from the PTO layer can be determined qualitatively by subtracting the intensity of the opposite Friedel pair disks due to the violation of Friedel's law³⁰⁻³². This polarization mapping is slightly different from strain mapping where the accurate position of the diffracted disks is calculated³⁵. The method for subtracting the intensities of opposite Friedel's pair disks would also work for analogous $Pb_xZr_{1-x}TiO_3$ (PZT) where polarization is suppressed under large tetragonality³⁶. The (hkl) Friedel pairs were chosen along $[001]_o$ and $[1\overline{1}0]_o$ directions to determine pure lateral and axial polarization respectively. We note that the other (hkl) directions may also break Friedel's law, but the associated polarization will be a combination of lateral and axial component. The polarization maps corresponding to regions delimited by rectangles in Figure 2b are shown in Figure 2d-e.

Question 2. In fact, 4D-STEM only maps the tetragonality (strain). Although it is usually

positively correlated with the polarization, but sometimes it is not. For example, at the surface of PZT, the polarization is suppressed although the tetragonality (strain) is larger (Nature Communications | 7:11318 | DOI: 10.1038/ncomms11318). In the vortex system, I would expect this case is more complicated. I wonder how the author handle this. Related discussion is needed to avoid any misleading.

Response: We agree with the reviewer that the polarization and strain do not always go hand in hand. However, in 4D STEM , we determine the strain by the change in the position of the disks relative to the parent materials. The polarity is associated with the relative change in the intensity of the opposite Friedel pair disks, that measured in our current work. We have now added the discussion in main manuscript as shown in previous question. To help with the discussion, we have also cited the paper mentioned by the reviewer.

Question 3. Vortex is a three-dimensional polar structure. Obviously, in the STEM image the polarization along the viewing direction is not uniform for this sample, i.e., near the core and far away from the core should have different polarization. For the conventional HAADF-STEM method without the depth resolution, it is very difficult to precisely measure the in-plane polarization to extract the chirality information. However, I do not see how the 4D-STEM solve this problem. Did the authors simply ignore this and extract the averaged polarization information? Detailed discussion is needed.

Response: We thank the reviewer for pointing this aspect. Indeed, the polarization varies near and away from the vortex core. Hence, the HAADF-STEM fails in this regard. However, 4D-STEM allows us to image the lateral and axial polarization simultaneously which is not possible the HAADF-STEM. We cannot access the axial polarization directly with accuracy in conventional HAADF-STEM. However, in 4D-STEM we can access the pure axial polarization because it has a different diffraction condition (different Friedel's diffraction pairs) than lateral polarization. Additionally we primarily observe polarization from the top half of polar vortices in PbTiO_3 layer due to the limited depth resolution (<http://arxiv.org/abs/2012.04134>). Hence, the lateral polarization do not cancel out one another. We can't measure polarization quantitatively, but our qualitative measurements are good enough for the present purposes studying domain boundaries and triple points). We illustrate this better in the following diagram pasted paste in a figure from a previous paper (<https://www.nature.com/articles/s41586-019-1092-8>) where we can measure the topological polarization texture buried underneath SrTiO_3 .

Figure: **a, b**, Reversed Ti-displacement vector map (top) based on the atomically resolved plane-view HAADF-STEM image (bottom) of a single skyrmion bubble (marked by a white circle in Extended Data Fig. 6a), showing the hedgehog-like skyrmion structure. The sketch of the superlattice in **b** is overlaid with the planar-view dark-field TEM image and gives a top view of the superlattice. **c**, Ti-displacement vector map (front) based on the atomically resolved cross-sectional HAADF-STEM image (back), showing a cylindrical domain with anti-parallel (up-down) polarization. The sketch in **b** is overlaid with the cross-sectional dark-field TEM image and shows the cross sectional view of the superlattice. **d, e**, The 4D-STEM image of a $[(\text{PbTiO}_3)_{16}/(\text{SrTiO}_3)_{16}]_8$ superlattice gives the ADF image (**d**) and maps of polar order using the probability current flow (**e**), which were reconstructed from the same 4D dataset. **f, g**, Multislice simulations of the beam propagation through the model structure from Fig. 2 show the ADF image (**f**) and the probability current flow (**g**), which were analyzed using the same process as the experimental data. The signals are not simple projections, but weighted by electron beam channelling towards the middle of the skyrmion bubble, where the polarization exhibits a Bloch-like character.

We have also added this discussion in the main manuscript and modified the schematic in Figure 2.

. On the other hand, 4D-STEM allows us to collect a diffraction pattern at each probe position, which can then be used to create precise maps of physical quantities such as strain and polarization. **The diffraction pattern in 4D-STEM offers a unique advantage over HAADF-STEM in polar vortices because direction of polarization can be accurately measured.** For the present experiment, we performed 4D-STEM imaging on a trilayer STO/PTO/STO along the $[110]_o$ zone axis (Figure 2a). We used a probe size of $\sim 7 \text{ \AA}$, larger than the STO/PTO unit cell dimensions ($\sim 4 \text{ \AA}$) to remove the atomic-resolution signal and to estimate the polarization at unit cell resolution.

Question 4. I wonder how to determine the position of vortex cores from the BF-STEM image in Fig. 1a, I can only distinguish the dark contrast of the wave shape. How to correlate the bright/dark contrast with the core positions. Besides, it would be better to remove a few red circles to show more details.

Response: BF-STEM is more sensitive to the diffraction contrast. Vortex features are continuous rotation of polarization vectors and hence they are more visible at lower collection angles in BF-STEM mode. In the BF STEM mode, they usually appear as dark cross. For better clarity, we have removed some of the red circles as the reviewer was suggesting.

Question 5. Both the virtual dark field image and the actual dark field image in Fig. 1c are constructed from diffraction information. Why are these domain walls invisible in virtual dark field images? In addition, the dark field image in Fig. 1c need to clarify the selected diffraction point.

Response: Some of the vortex domain walls may be missing in the virtual dark field images because they do not capture the entire information. Vortex domain walls originate from the change in lateral and axial polarization which can only be determined by difference in the intensity of the opposite Friedel pair disks.

Here is an example showing all the dark field images taken from disk 1, 2, 3 and 4. Even, if we combine everything, we will still find a few boundaries missing. It is only the normalized subtraction of (1-2) and (3-4) that reveals all of the vortex domain boundaries.

Figure S4: Dark field images corresponding disks 1-4.

Question 6. Can different domain walls be distinguished by dark field images? e.g., through branching or stripe periodicity.

Response: No, they cannot be distinguished by just dark field images. This is because the vortex domain walls are *only* identified by change lateral and axial polarization. The lateral and axial polarization cannot be identified by just one dark field image.

We have now also included the virtual dark field image created from four different disks. We cannot solely identify all of the polarization domain boundaries from just one dark field image.

Question 7. I found some problems in the cited references in this manuscript. For example, the authors mentioned the “The first experimental demonstration was in 25, where chirality switches...”. In fact, at the same time another group published an article of switching chirality of polar vortex at the atomic scale STEM and dark-field TEM, which has been totally ignored in this manuscript. (Sci. China-Phys.Mech. Astron. 65, 237011 (2022), <https://doi.org/10.1007/s11433-021-1820-4>). Another one is the authors mentioned that “novel polarization textures in ferroelectrics such as merons, polar flux-closure domains, vortices.....in oxide superlattices”, ignoring the representative polar antivortex (Nature Communications | (2021) 12:2054, <https://doi.org/10.1038/s41467-021-22356-0>), and three-fold polar vertices (Nature Communications | (2022) 13:6340, <https://doi.org/10.1038/s41467-022-33973-8>)

Response: We thank reviewer for the suggestions. We have included all the citations in the introduction of our manuscript.

Question 8.. Some minor suggestions: Fig. 1c lacks the scale bar; Fig. 3a lacks the axis; “Once

the atoms were identified, the atomic planes were divided into different zone axis such as along $[001]_o$ and $[001]_o$ ” seems to be a typo here.

Response: We have corrected the typo and added the scale bar in Figure 1c.

Reviewer #2 (Remarks to the Author):

S. Susarla et al. reported the chirality engineering of the topological polar vortex via atomic-scale symmetry-breaking operations. In this work, 4D-TEM results display the topology-driven three-dimensional domain walls, and the manuscript was well organized. As the author said, the chirality of the polar vortex is governed by the perpendicular and parallel polarization of the tubular vortex. In my opinion, polarization vector mapping of the vortex region in in-plane geometry will be a more intuitive way to demonstrate the vortex' chirality. Furthermore, the 4D-TEM data needs to be reanalyzed, otherwise it is difficult to support the existing conclusions. Two main concerns are as follows:

Response: We thank the reviewer for the comments. However the reviewer has not completely understood our motivation behind performing 4D-STEM. Polar vortices have both axial and lateral polarization components, out of which the former cannot be directly measured via HAADF-STEM due to the projection problem. Since 4D-STEM uses two different Friedel pair disks to map out the lateral and axial polarization, it is more intuitive to use 4D-STEM to quantify the polarization. A more detailed description is given in the latter responses.

Question 1. In Figure 1b, it is difficult for the reader to locate the vortex center, which is consistent with the position marked by the author.

Response: We have now reanalyzed Figure 1b which is shown below:

Figure S1: Polarization vector maps overlaid on the drift-corrected HAADF-STEM images. The yellow vector indicates the direction of polarization. The underlying red/blue contrast is the curl of the displacement.

We want to clarify that we are marking the vortex center at the maxima/ minima point of the polarization curl which is defined as below:

$$\theta = \frac{1}{2} \left(\frac{\partial u}{\partial y} - \frac{\partial v}{\partial x} \right)$$

This is how we have marked the center position of the vortices. We have put the zoomed-out version of the polarization maps, and their corresponding in-plane and out-of-plane strain maps in the supplementary information.

Figure S2: Strain maps (top two) and infinitesimal rotation (bottom) extracted by A-site fitting of atoms in Figure S1.

Question 2. The combination of Figure 2 a, 2e and Figure 4e, there is a dislocation structure with respect to the tubular vortex across the γ -domain wall, in accordance with Figure 1b shows. However, in Fig. S3 and S4, the Virtual image has obviously the same boundary (in the middle of the image) as in Fig.2b, but the author does not identify it as any vortex domain wall. Could authors explain this difference? Perhaps, the polarization vector mapping on HAADF-STEM image will illustrate the issue more directly.

Response: We thank the reviewer for pointing out this aspect. We have now reanalyzed the 4D STEM datasets where we have included the missing vortex domain wall that reviewer pointed

out.

Figure S6: Virtual image, polarization, and helicity maps from different 4D STEM datasets showing the repeatability of different chiral/achiral boundaries in the PTO/STO trilayer. The presence of triple point topologies is evident whenever the chiral and achiral boundaries intersect one another. Scale bar: 20 nm for all panels.

Figure S7: Virtual image, polarization, and helicity maps from different 4D STEM datasets showing the repeatability of different chiral/achiral boundaries in the PTO/STO trilayer. The presence of triple point topologies is evident whenever the chiral and achiral boundaries intersect one another. Scale bar: 30 nm

The polarization maps on the in-plane HAADF-STEM image will not reveal all types of domain boundaries as the axial polarization cannot be measured directly in the HAADF-STEM.

Reviewer #3 (Remarks to the Author):

The manuscript by Susarla et al. presents a structural analysis of the 3D domain wall network in topological polar vortices formed in (SrTiO₃)₁₆/(PbTiO₃)₁₆/(SrTiO₃)₁₆ trilayers grown on SrRuO₃-buffered DyScO₃ substrates. The microstructure of the films is examined in detail by means of 4D-STEM imaging. Thus, lateral, and axial polarization maps are obtained by taking the normalized intensity difference between opposite Friedel pair disks. This allows identifying three distinct types of domain wall configurations having different parallel/antiparallel axial/lateral components. Thus, two chiral and one achiral domain walls are identified. Finally, the authors observe that the domain walls meet at triple points, which typically appear in pairs. They hypothesize that these topological defects could lead to unique electrostatic and magnetic properties useful for quantum sensor applications.

The manuscript presents an original and very good experimental and analytical work on the microstructure of domain walls in topological polar vortices. Overall, the manuscript is clearly written, and the figures are well elaborated. I enjoyed reading the manuscript, although I suggest introducing a couple of changes to make it a bit more comprehensible. In addition, I am listing also here some minor amendments.

Response: We thank the reviewer for appreciating our manuscript. We have answered almost all of the reviewer questions.

Question 1. In page 4 of the manuscript, "Supplementary Information" should be deleted, as there is no additional information in the SI referring to the trilayer stack.

Response: This has now been deleted.

Question 2. At the end of page 5, it reads that in Figure 1b "a zig-zag type pattern, giving rise to a net in-plane polarization rotation along [001]_o (lateral component) indicated as P_x". I have difficulties seeing this net in-plane polarization. Could the authors plot the resulting in-plane polarization by averaging it along the horizontal direction and plotting it next to the image? This is not obvious from the figure. It seems to me that the P_x at the top of the PTO film should cancel with the P_x at the bottom of the PTO film.

Response: We thank the reviewer for noticing this aspect. We have re-plotted our polarization vector maps in Figure 1b (i) where we observe that the top portion of lateral polarization is not equivalent with the bottom portion of the polarization. Additionally, we have a schematic in

Figure 1b (ii) explaining the origin of *net* lateral polarization.

Question 3. Again, in Figure 1b, it seems there are vortex cores also in the bottom STO layer. Can the authors comment on this?

Response: We apologize to reviewers for this mistake. We had some errors in fitting the A sites. We re-did the A site gaussian fitting and replaced the earlier images better one as new Figure 1b. Refer to our response to reviewer 2 Question#1

Question 4. The scale is missing in Figure 1c.

Response: The scale bar has been added now.

Question 5. It is a bit complicated to follow the discussion of the possible triple points depicted in Figure 3b. Could the authors correlate the triple points observed in Figure 3a (and also in the SI) to the triple point pairs represented in Figure 3b? This would make it easier to follow the explanation.

Response: We thank the reviewers for the response. We have now included the labels in Figure 3b and also indicated their corresponding location in Figure 3a.

Question 6. In the caption of Figure 4 it reads "The chirality for each type of domain is indicated by a sketched hand", but in the figure there are no sketched hands. Please, remove the sentence.

Response: We have removed that sentence.

Question 7. In the Materials and Methods the authors should indicate how was the sample for S/TEM prepared.

Response: We have now added the STEM sample preparation part in Materials and Methods.

STEM Sample preparation: In-plane $[(\text{PbTiO}_3)_{16}/(\text{SrTiO}_3)_{16}]$ trilayer grown on $\text{SrRuO}_3/\text{DyScO}_3$ substrate were mechanically polished using a 0.5° wedge in Allied Multiprep. The samples were subsequently Ar ion milled in a Gatan Precision Ion Milling System, starting from 3.5 keV at 4° down to 1 keV at 1° for the final polish. The HAADF-STEM images were acquired using double aberration corrected TEAM I microscope operated at 300 kV under non-monochromated mode.

Question 8. In page 14, please change "low-pass and high-pass Gaussian filters" for "band-pass Gaussian filters".

Response: We have changed the typos

Based on my previous comments I recommend the publication of the manuscript of Susarla et al. in Nat. Commun. after minor revisions.

REVIEWERS' COMMENTS

Reviewer #1 (Remarks to the Author):

The revised manuscript addresses my concerns regarding the depth resolution problem in 4DSTEM and polarization information extraction, the authors state that the signal primarily originates from the top half of polar vortices in the PbTiO₃ layer, as supported by simulation results (<http://arxiv.org/abs/2012.04134>). The virtual imaging ability and the ability to obtain pure axial polarization in 4DSTEM is indeed a valuable advantage over conventional STEM imaging. I understand that while quantitative measurements of polarization may not be feasible, the qualitative measurements in this work still provide valuable insights into studying domain boundaries and triple points. Overall, I'm happy with the response to all the questions and comments. Thus, I recommend the publication of this manuscript in NC.

Reviewer #2 (Remarks to the Author):

The authors have addressed my concerns.

Reviewer #3 (Remarks to the Author):

The authors have satisfactorily addressed most of my concerns and I therefore suggest publication of the manuscript by Susarla et al. in Nature Communications.